# Hirshfeld Surface Analysis and Density Functional Theory Calculations of 2-Benzyloxy-1,2,4-triazolo[1,5-*a*] quinazolin-5(4*H*)-one: A Comprehensive Study on Crystal Structure, Intermolecular Interactions, and Electronic Properties

**Ahmed H. Bakheit** *, **Hatem A. Abuelizz**  **and Rashad Al-Salahi** * 



Department of Pharmaceutical Chemistry, College of Pharmacy, King Saud University, Riyadh 11451, Saudi Arabia; habuelizz@ksu.edu.sa
* Correspondence: abakheit@ksu.edu.sa (A.H.B.); ralsalahi@ksu.edu.sa (R.A.-S.)

**Abstract:** This study employs a comprehensive computational analysis of the 2-benzyloxy-1,2,4-triazolo[1,5-*a*] quinazolin-5(4*H*)-one (ID code: CCDC 834498) to explore its intermolecular interactions, surface characteristics, and crystal structure. Utilizing the Hirshfeld surface technique and Crystal Explorer 17.5, the study maps the Hirshfeld surfaces for a detailed understanding of atom pair close contacts and interaction types. The study also investigates the compound's electronic and optical characteristics using Frontier Molecular Orbital (FMO) analysis and Global Reactivity Parameters (GRPs). The compound is identified as electron-rich with strong electron-donating and accepting potential, indicating its reactivity and stability. Its band gap suggests Nonlinear Optical (NLO) attributes. The Molecular Electrostatic Potential (MEP) map reveals charge distribution across the compound's surface. The computational methods' reliability is validated by the low Mean Absolute Error (MAE) and Mean Squared Error (MSE) in the comparison of experimental and theoretical bond lengths and angles.

**Keywords:** triazoloquinazoline; DFT; Hirshfeld study; crystal structure

## 1. Introduction

The class of heterocyclic compounds known as 1,2,4-triazolo[1,5-*a*]quinazolines has recently emerged as being of critical importance in the fields of medicinal chemistry and the creation of new drugs. These compounds possess the ability of a wide variety of biological functions and have demonstrated promise as potential therapeutic agents in a number of different disease categories [1]. Their pharmacological activities have shown evidence of a variety of different pharmacological activities, such as adenosine antagonist, anticancer, antiviral, antibacterial, antifungal, and anti-inflammatory behaviors [2,3]. Because of their adaptability, they are appealing as potential targets for the creation of new therapies. The high reactivity of dibenzyl-*N*-cyanoimidocarbonate towards 2-hydrazinobenzoic acid resulted in the formation of corresponding 1,2,4-triazoles as the primary condensation product, which, in turn, produced the target 2-benzyloxy-1,2,4-triazolo[1,5-*a*]quinazolin-5-one that was created through the process of intramolecular condensation [4]. The vast majority of triazoloquinazoline analogs include an inherent lactam/thiolactam group, and the chemistry that was performed on this functional group enabled researchers to obtain access to a wide variety of heterocyclic compounds [5,6]. Moreover, the triazoloquinazoline system combined the chemical characteristics of quinazoline and triazole, which made it suitable for a wide variety of nucleophilic and electrophilic substitution reactions on either the triazole ring or the quinazoline component of the molecule [1].

The target 2-benzyloxy-[1,2,4]triazolo[1,5-*a*]quinazoline was meticulously investigated using spectroscopic methods, culminating in the confirmation of the crystal structure through

X-ray diffraction studies. In continuation of our study on triazoloquinazoline chemistry, particularly studying the crystal structure of 1,2,4-triazolo[1,5-a]quinazoline derivatives.

The current research provides an in-depth study of the intimate intermolecular contacts amongst the molecules via Hirshfeld surface analysis, a valuable tool to quantify the interactions embedded within the crystal structure. The energies associated with molecular interactions and the lattice were calculated using 3D energy frameworks [7–9]. In addition, density functional theory calculations were conducted to illuminate the compound's electronic and chemical attributes. Studies of the MEP map surface were also performed, aiming to identify the chemically reactive sites on the molecular surface of the 2-benzyloxy-1,2,4-triazolo[1,5-a]-quinazoline molecule.

## 2. Materials and Methods

### 2.1. Hirshfeld Surface Studies, Interaction Energies and 3D Energy Frameworks

Hirshfeld Surface Analysis (HSA) [10,11] serves as an effective method for examining intermolecular interactions within a crystal structure. This technique is unique in its ability to calculate and visually depict these interactions, with each crystal structure producing unique outcomes. The analysis is carried out by entering the crystallographic information file (CIF) of the molecule under investigation into the Crystal Explorer 17.5 [12] software. The CIF (CCDC 834498) used in this study was obtained from the previous work of one of the authors.

One of the primary features of HSA is the generation of 3D molecular surface contours and 2D fingerprint plots. These plots and contours form a van der Waals (vdW) surface around the molecule, representing the space that the molecule occupies within the crystal structure. Importantly, contact distances from points on this surface to atoms within (di) and outside (de) the surface are determined by the various vdW radii of the atoms. These distances can then be normalized (dnorm) [13].

This technique provides a detailed view of intermolecular interactions, varying from short to long, in a crystal structure, which are typically induced by hydrogen bond donors and acceptors. The visualization of these interactions is represented by color-coded fingerprint plots and contour surfaces, with distances that are shorter or longer than the sum of the vdW radii being depicted in a range from red (shorter) to white and blue (longer). For the title compound, Hirshfeld surface plots and fingerprint plots have been generated using Crystal Explorer 17.5 [10,12,14] software.

In this study, a detailed analysis was carried out to understand the molecular interaction energies within the crystal structure. To perform this, the monomer wave functions were used as a starting point, and the CE-B3LYP/6–31G(d,p) method was employed [9]. This method was particularly chosen because it is essential for conducting 3D energy framework studies, which provide valuable insights into the characteristics and behavior of the molecule within the crystal structure.

The computation process involved the calculation of different types of molecular interaction energies. These include electrostatic energy, which deals with the forces between charged particles; polarization energy, which concerns the interactions caused by the distortion of a molecule's electron cloud by other nearby charge distributions; dispersive energy, which accounts for the weak attractive forces resulting from temporary fluctuations in a molecule's electron distribution; and repulsive energy, which is the energy required to overcome the forces that prevent two molecules from occupying the same space. Adding these different types of interaction energies together provides the total interaction energy of the molecule under investigation, as depicted in the following formula:

$$E_{tot} = E_{ele} + E_{pol} + E_{dis} + E_{rep} \qquad (1)$$

To account for potential variations in the molecular energies obtained from the generated wave function using density functional theory, scale factors (represented by K's in the equation) are applied:

$$E_{tot} = K_{ele}E'_{ele} + K_{pol}E'_{pol} + K_{dis}E'_{dis} + K_{rep}E'_{rep} \tag{2}$$

The interaction energy breakdown method employed in this study has been extensively applied in energy-decomposition procedures via both variational and perturbation-based methodologies [9]. The classical electrostatic energy of interaction between monomer charge distributions, $E'_{ele}$, and the exchange–repulsion energy, $E'_{rep}$, were derived from the antisymmetric product of the monomer spin orbitals, as per the method specified. The polarization energy, $E'_{pol}$, was calculated by summing over atoms with terms like $\frac{1}{-2\alpha}|F|^2$, where the electric field F, computed at each atomic nucleus from the charge distribution of the other monomer, represented isotropic atomic polarizabilities. The dispersion energy term, $E'_{dis}$, was obtained by summing Grimme's D2 [15] dispersion correction across all intermolecular atom pairs. The scale factors, such as $K_{ele}$, as mentioned in Equation (1), are determined through calibration against results derived from quantum mechanics. [9,16]

The calculated interaction energies were then utilized to create 3D energy frameworks. These frameworks are instrumental in providing a visual understanding of how the tested molecules are arranged within their respective crystal structures [9,11,17].

*2.2. Density Functional Theory Calculations*

Investigations into the structure at the molecular level and computations of electronic properties were performed using a variety of computational methodologies. One such method is DFT, a theoretical framework that analyses the distribution function of electron density. The process of structural optimization was executed in a gas phase environment utilizing the DFT/B3LYP hybrid functional along with a 6-311G(d,p) basis set. This operation was carried out using Gaussian 09W [18], a popular computational chemistry software. Additionally, the long-range correction functional wB97XD, a component of DFT, was employed for more accurate results. The outcomes of these computations were visualized using GaussView 6.0 [19], a graphical interface used with Gaussian. The energies of the highest occupied molecular orbital (HOMO) and the lowest unoccupied molecular orbital (LUMO) were determined, and these values were used to calculate global reactivity descriptors following Koopman's theorem [20]. Furthermore, second-order perturbation theory, a method used to approximate the exact solution of a problem, was applied to compute the interaction energies through Fock matrix analysis. The Fock matrix is a key concept in Hartree-Fock theory, which is a method of approximation for the determination of the wave function and the energy of a quantum many-body system in a stationary state. Moreover, the potential electrophilic and nucleophilic reactive sites on the molecular surface were pinpointed by creating and analyzing MEP maps. These maps are graphical representations of the electrostatic potential at the electron cloud surface and are used to visualize areas of chemical reactivity.

### 3. Results and Discussion
*3.1. The Crystal Structure Data and Refinement Details (CCDC 834498)*

The target 2-benzyloxy-1,2,4-triazolo[1,5-*a*]-quinazolin-5(4*H*)-one (Scheme 1) was previously synthesized and fully described [3]. The Diamond 5 Preview diagram [21] of the title compound, 2-benzyloxy-1,2,4-triazolo[1,5-*a*]quinazolin-5-one with ball and stick drawn at a 50% probability is shown in Figures 1 and 2. The empirical formula of the compound is $C_{16}H_{12}N_4O_2$, which gives it a formula weight of 292. The crystal structure was determined at a temperature of 153(2) K. The crystal system for this compound is monoclinic, and the space group is P21/n. The unit cell parameters are: a = 5.0319(15) Å, b = 28.207(9) Å, c = 9.408(3) Å. The angles of the unit cell are $\alpha = 90.00°$, $\beta = 99.503(5)°$, $\gamma = 90.00°$. The volume of the unit cell is 1317.0(7) Å$^3$ [6]. There are four formula units

(Z = 4) in the unit cell. The calculated density of the crystal is 1.474 g/cm. The absorption coefficient, μ, is 0.102 mm$^{-1}$.

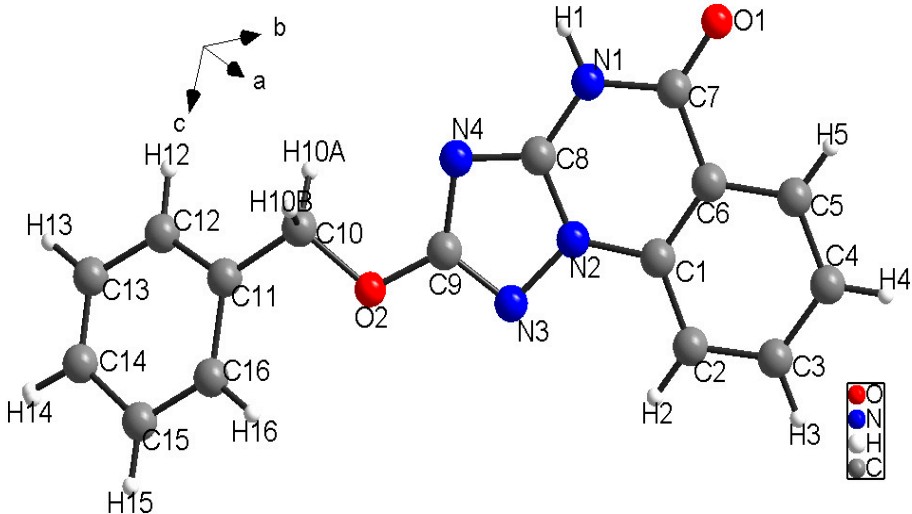

**Scheme 1.** Synthetic routes for 2-benzyloxy-1,2,4-triazolo[1,5-*a*]quinazolin-5(4*H*)-one.

**Figure 1.** Diamond plots for 2-Benzyloxy-1,2,4-triazolo[1,5-a]-quinazolin-5(4H)-one asymmetric units, aligned with crystallographic directions.

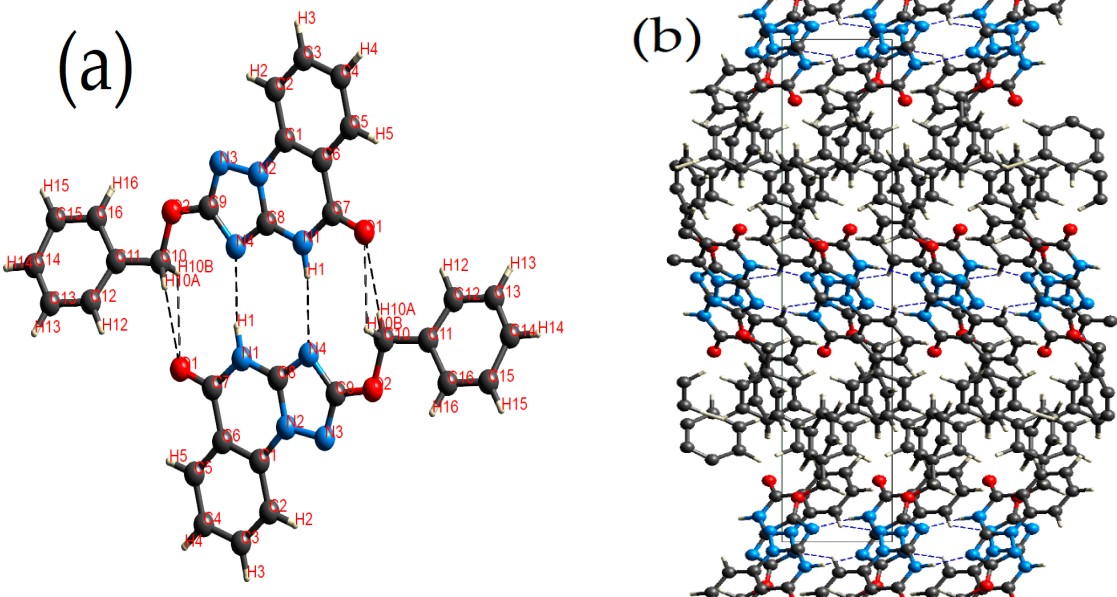

**Figure 2.** Crystal packing of C$_{16}$H$_{12}$N$_4$O$_2$ (I) (**a**) dimers and (**b**) polymers viewed along the c-axis. Dashed lines indicate classical hydrogen bonds N1-H1···N4 intermolecular interactions forming chains along the c-axis.

The crystal was measured using MoKα radiation (λ = 0.71073 Å), and the crystal size was 0.5 × 0.1 × 0.03 mm. The diffraction data was collected over a 2θ range of 5.78° to 54°. A total of 8172 reflections were collected, with 2849 of them being independent ($R_{int}$ = 0.0532, $R_{sigma}$ = 0.0903). The data/restraints/parameters ratio is 2849/6/The goodness-of-fit on $F^2$ is 0. For reflections with I ≥ 2σ(I), the final R indexes are $R_1$ = 0.0470, $wR_2$ = 0. Including all data, the final R indexes are $R_1$ = 0.0930, $wR_2$ = 0. The largest difference in the electron density map is 0.20 e Å$^{-3}$ for a peak and −0.20 e Å$^{-3}$ for a hole. This gives an indication of the quality of the fit between the observed and calculated electron density.

### 3.2. Geometry Optimizations

The molecule being studied, 2-benzyloxy-1,2,4-triazolo[1,5-*a*]quinazolin-5(4*H*)-one, consists of three connected rings: a benzene ring, a triazol ring, and a central heterocyclic quinazolinone ring. The benzyloxy substituent also forms an additional ring system. These rings can adopt various conformations, and the geometry optimization results you've provided indicate the preferred conformation.

The results show that both the pure and substituted forms of the three rings in the molecule energetically favor a planar conformation. This is reflected in the dihedral angles of 0.0° for all the rings. A dihedral angle of 0.0° indicates that the atoms are co-planar, meaning they lie in the same plane, demonstrating a flat structure. This planarity is typical for aromatic and heterocyclic systems where electron delocalization contributes to stability.

The quinazolinone core of the molecule, a heterocyclic ring, can be divided into three distinct facets, denoted as (C8, N4, C9, N3, N2), (C8, N2, C1, C6, C7, N1), and (C1, C2, C3, C4, C5, C6). Each of these facets displays a dihedral angle of 0.0°, corroborating their planar configuration. In contrast, the attached benzene ring, denoted as D(C11, C12, C13, C14, C15, C16), presents a dihedral angle of 33.404(93)° (Figure 3). This non-zero angle implies a deviation from planarity, suggesting a different geometric structure for this ring compared to the quinazolinone core.

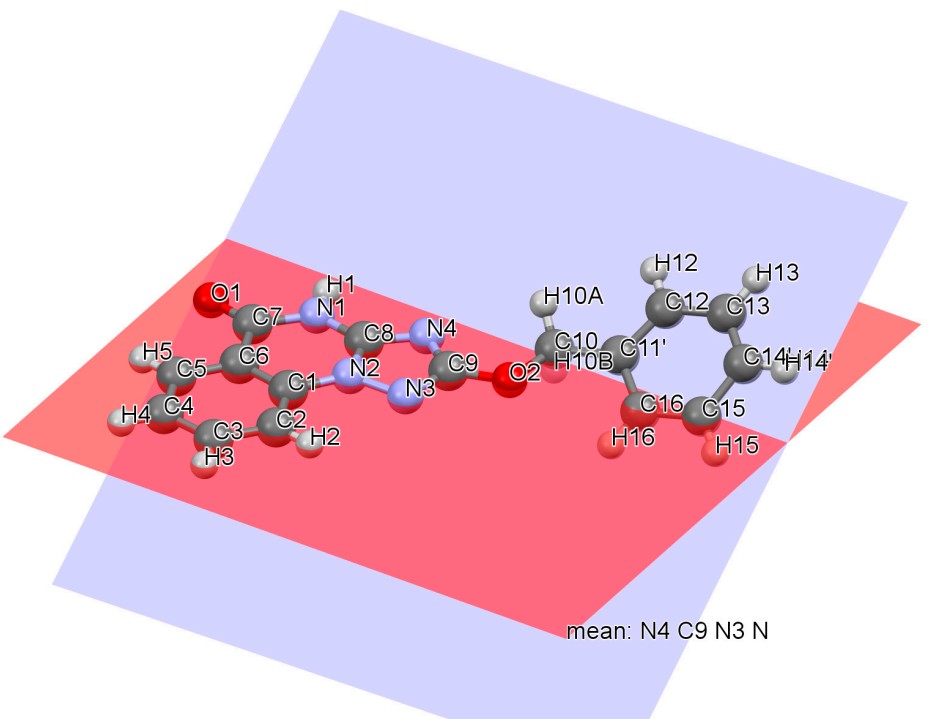

**Figure 3.** Illustrates the centroids of the rings: Cg1 (highlighted in red) includes the atoms (C8, N4, C9, N3, N2), (C8, N2, C1, C6, C7,N1), and (C1, C2, C3, C4, C5, C6); whereas Cg2 (depicted in blue) encompasses the atoms C11, C12, C13, C14, C15, C16.

The results, therefore, suggest that the benzene, triazol, and quinazolinone rings, as well as the two sides of the central heterocyclic ring, all adopt a planar conformation in the 2-benzyloxy-1,2,4-triazolo[1,5-*a*]quinazoline molecule. This planarity can influence various molecular properties and behaviors, such as reactivity and interactions with other molecules, which can be crucial in contexts like drug-receptor interactions.

While the DFT optimization provides helpful insights into the preferred molecular conformation, it's also important to validate these theoretical results with experimental data, such as from single-crystal X-ray diffraction studies, to ensure a comprehensive understanding of the molecule's geometry in its ground state. The information provided pertains to the comparison of computationally refined molecular geometries with experimental models for a given compound. These comparisons are detailed in Tables 1 and 2. The refined computational structures align well with experimentally reported structures, suggesting that the computational methods used are reliable for modeling the molecular geometry of the studied compound. This alignment was assessed by comparing key computed and experimental bond lengths and bond angles, as detailed in Tables 1 and 2.

**Table 1.** A comparison of bond lengths (in Å) for the title compounds, as obtained from experimental X-ray measurements and theoretical calculations using the B3LYP/6-311G(d,p) method.

| Atom | Length/Å | | Absolute Error (AE) | Square Absolute Error (SAE) |
|---|---|---|---|---|
| | SC-XRD | DFT | | |
| O1—C7 | 1.223(2) | 1.258 | 0.035 | 0.001 |
| O2—C9 | 1.339(2) | 1.43 | 0.091 | 0.008 |
| O2—C10 | 1.450(2) | 1.43 | 0.02 | 0 |
| N1—C7 | 1.397(2) | 1.473 | 0.076 | 0.006 |
| N1—C8 | 1.364(2) | 1.342 | 0.022 | 0 |
| N2—N3 | 1.391(2) | 1.399 | 0.008 | 0 |
| N2—C1 | 1.396(2) | 1.467 | 0.071 | 0.005 |
| N2—C8 | 1.351(2) | 1.336 | 0.015 | 0 |
| N3—C9 | 1.320(2) | 1.296 | 0.024 | 0.001 |
| N4—C8 | 1.333(2) | 1.293 | 0.04 | 0.002 |
| N4—C9 | 1.380(2) | 1.473 | 0.093 | 0.009 |
| C1—C2 | 1.394(3) | 1.4 | 0.006 | 0 |
| C1—C6 | 1.399(3) | 1.394 | 0.005 | 0 |
| C2—C3 | 1.381(3) | 1.403 | 0.022 | 0 |
| C3—C4 | 1.392(3) | 1.405 | 0.013 | 0 |
| C4—C5 | 1.387(3) | 1.403 | 0.016 | 0 |
| C5—C6 | 1.401(3) | 1.4 | 0.001 | 0 |
| C6—C7 | 1.479(3) | 1.539 | 0.06 | 0.004 |
| C10—C11 | 1.500(3) | 1.54 | 0.04 | 0.002 |
| C11—C12 | 1.379(5) | 1.401 | 0.022 | 0.001 |
| C11—C16 | 1.465(4) | 1.401 | 0.064 | 0.004 |
| C12—C13 | 1.385(6) | 1.401 | 0.016 | 0 |
| C13—C14 | 1.281(5) | 1.401 | 0.12 | 0.014 |
| C14—C15 | 1.435(5) | 1.401 | 0.034 | 0.001 |
| C15—C16 | 1.393(6) | 1.401 | 0.008 | 0 |
| Mean | | | 0.037 | 0.002 |

**Table 2.** Comparison of bond angles (in degrees) for the title compounds, drawn from both experimental X-ray findings and theoretical calculations using the B3LYP/6-311G(d,p) methodology.

| Atoms | SC-XRD | DFT | Absolute Error (AE) | Square Absolute Error (SAE) |
|---|---|---|---|---|
| | | | Angle/° | |
| C9—O2—C10 | 115.94(15) | 109.471 | 6.469 | 41.845 |
| C8—N1—C7 | 122.56(16) | 118.776 | 3.784 | 14.317 |
| N3—N2—C1 | 125.89(16) | 126.335 | 0.445 | 0.198 |
| C8—N2—N3 | 109.78(16) | 110.4 | 0.62 | 0.384 |
| C8—N2—C1 | 124.32(16) | 123.265 | 1.055 | 1.113 |
| C9—N3—N2 | 100.65(15) | 104.677 | 4.027 | 16.214 |
| C8—N4—C9 | 100.89(16) | 104.738 | 3.848 | 14.806 |
| N2—C1—C6 | 116.03(17) | 117.84 | 1.81 | 3.277 |
| C2—C1—N2 | 122.16(18) | 121.748 | 0.412 | 0.17 |
| C2—C1—C6 | 121.80(19) | 120.412 | 1.388 | 1.928 |
| C3—C2—C1 | 118.12(19) | 119.567 | 1.447 | 2.095 |
| C2—C3—C4 | 121.51(19) | 120.038 | 1.472 | 2.167 |
| C5—C4—C3 | 119.9(2) | 120.047 | 0.147 | 0.022 |
| C4—C5—C6 | 120.09(19) | 119.586 | 0.504 | 0.254 |
| C1—C6—C5 | 118.57(18) | 120.35 | 1.78 | 3.169 |
| C1—C6—C7 | 121.65(18) | 117.977 | 3.673 | 13.49 |
| C5—C6—C7 | 119.78(18) | 121.673 | 1.893 | 3.583 |
| O1—C7—N1 | 120.98(18) | 120.867 | 0.113 | 0.013 |
| O1—C7—C6 | 123.46(19) | 120.867 | 2.593 | 6.726 |
| N1—C7—C6 | 115.55(17) | 118.267 | 2.717 | 7.38 |
| N2—C8—N1 | 119.84(18) | 123.875 | 4.035 | 16.28 |
| N4—C8—N1 | 128.80(17) | 125.998 | 2.802 | 7.849 |
| N4—C8—N2 | 111.34(16) | 110.127 | 1.213 | 1.472 |
| O2—C9—N4 | 124.51(17) | 124.971 | 0.461 | 0.212 |
| N3—C9—O2 | 118.13(17) | 124.971 | 6.841 | 46.794 |
| N3—C9—N4 | 117.35(18) | 110.059 | 7.291 | 53.162 |
| O2—C10—C11 | 108.60(16) | 109.471 | 0.871 | 0.759 |
| C12—C11—C10 | 122.8(2) | 120 | 2.8 | 7.84 |
| C12—C11—C16 | 114.3(3) | 120 | 5.7 | 32.49 |
| C16—C11—C10 | 122.9(2) | 120 | 2.9 | 8.41 |
| C11—C12—C13 | 124.5(4) | 120 | 4.5 | 20.25 |
| C14—C13—C12 | 119.5(4) | 120 | 0.5 | 0.25 |
| C13—C14—C15 | 123.7(3) | 120 | 3.7 | 13.69 |
| C16—C15—C14 | 116.6(4) | 120 | 3.4 | 11.56 |
| C15—C16—C11 | 121.4(4) | 120 | 1.4 | 1.96 |
| Mean | | | 2.532 | 10.175 |

### 3.2.1. Bond Lengths

Table 1 contains a comparison of experimental and theoretical bond lengths. The compound under study shows a good agreement between calculated and experimental values, highlighting the reliability of the theoretical calculations. The Mean Absolute Error (MAE) is 0.037 Å, and the Mean Squared Error (MSE) is 0.002 Å. The MAE provides a measure of the average magnitude of the errors between the predicted and observed values without considering their direction. A smaller MAE indicates a better fit of the model to

the data. In this case, an MAE of 0.037 Å indicates that, on average, the predicted bond lengths are within 0.037 Å of the experimentally determined values, which suggests a good fit. The MSE is another measure of accuracy, which squares the errors before averaging them, thus giving more weight to larger errors. A smaller MSE indicates a better fit of the model to the data. Here, an MSE of 0.002 Å$^2$ also suggests a good fit.

### 3.2.2. Bond Angles

Table 2 provides a comparison of experimental and theoretical bond angles. Again, there is a good match between the calculated and experimental data, with an MAE of 2.532° and an MSE of 10.175°. The relatively small MAE of 2.532° tells us that, on average, the predicted bond angles are within 2.532° of the observed values. The MSE of 10.175°$^2$, while larger, is still indicative of accuracy given the squared nature of this metric.

### 3.3. *Hirshfeld Surface Analysis*

Hirshfeld surface analysis provides insight into intermolecular interactions in the crystal state by characterizing the electron density associated with molecular contact points [12,15,22]. The Hirshfeld surface encloses a molecule and is defined by points where the electron density from the molecule of interest equals that contributed by neighboring molecules [10,13,23]. Hirshfeld surfaces were generated for the title compound using Crystal Explorer 17.5 and mapped with de, di, dnorm, shape index, curvedness, and two-dimensional fingerprint plots [10,12]. Figure 4 depicts the Hirshfeld surface maps for de, di, dnorm, shape index fragment, and curvedness of the molecule. Fingerprint plots can be decomposed to highlight specific atom-atom contacts, allowing separation of interaction types that would otherwise overlap [8,15,24,25]. This decomposition provides insight into the relative contributions of different intermolecular contacts to the crystal packing [13,26–28].

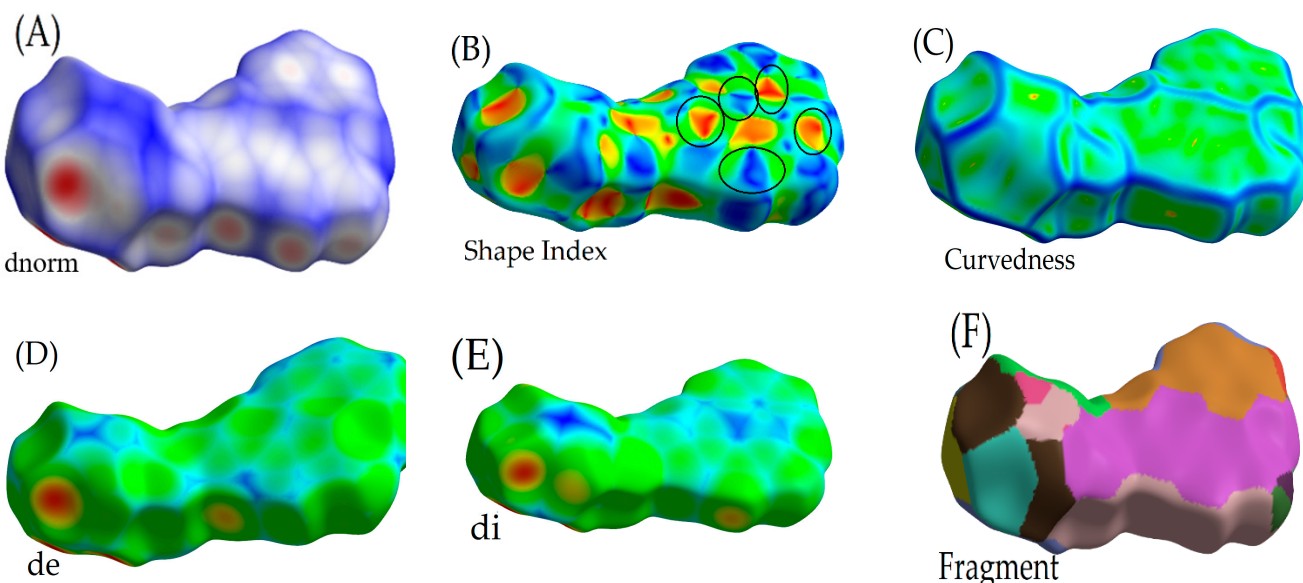

**Figure 4.** Hirshfeld surfaces mapped with (**A**) dnorm, (**B**) shape index surfaces showing $\pi\cdots\pi$ interactions, with red and blue triangles within a black ellipse indicating bumps and hollow regions, respectively, evidencing the $\pi\cdots\pi$ stacking area on the Hirshfeld surfaces, (**C**) curvedness with flat areas emphasizing the ring contributions in $\pi$-stacking interactions, (**D**) de, (**E**) di, and (**F**) fragment of the title molecule.

Highlighted on the molecular Hirshfeld surface in Figure 4 are various contacts, mapped using standard indices such as dnorm, shape index, curvedness, di, de, and fragment patches. The colored regions on these surfaces facilitate the analysis of diverse molecular surface properties. Specifically, red and blue regions on the dnorm represent shorter

and longer inter-contacts, respectively, while white indicates contacts approximating the van der Waals radii. Red regions correspond to negative potential with electrophilic characteristics, and blue ones to positive potential with nucleophilic characteristics. The shape index mapped on the Hirshfeld surface identifies red triangle concave regions as cyclic stacking interactions and blue triangle convex regions as ring atoms of the molecule [29]. Different colors of fragment patches represent molecular interactions across the molecular region. In Figure 4D,E, de and di denote distances from the Hirshfeld surface to the nearest nuclei outside and inside the surface, respectively. The volume within the Hirshfeld surface is computed to be 322.01 $Å^3$, with an area of 309.08 $Å^2$.

Figure 5 presents the Hirshfeld surfaces of $C_{16}H_{12}N_4O_2$, with dnorm mapped alongside neighboring molecules. Red spots on the surface indicate inter-contacts involved in intermolecular interaction [22], while blue areas signify regions too distant for neighboring atoms to interact [24]. The 3D dnorm surfaces are plotted over a fixed color scale of——1.0409 (red) to 1.0195 (blue), with a standard (high) surface resolution.

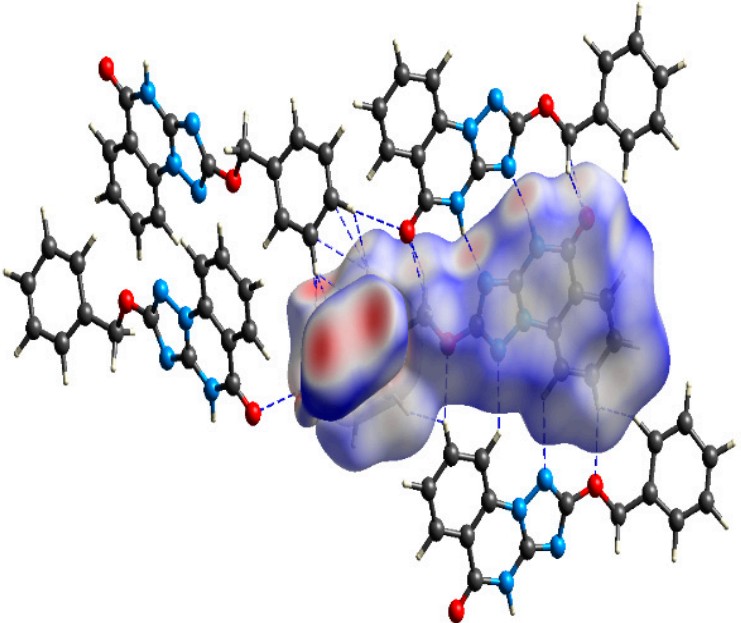

**Figure 5.** The red circular collapsing on the dnorm surface of the title compound structure represents the N—H···N and C—H···O intermolecular interactions.

The Hirshfeld surfaces, which are visual representations of molecular interactions, mapped over dnorm are illustrated in Figure 6. The primary interaction between oxygen (O) and nitrogen (N) with hydrogen (H) atoms is clearly evident in the region marked in red and blue, respectively. Other notable interactions represented on the Hirshfeld surfaces correspond to carbon-carbon (C–C) or hydrogen-hydrogen (H–H) contacts.

The subtle intermolecular interactions of the compound under focus are presented in Table 3, with the 2D&3D fingerprint plots depicted in Figures 7 and 8. The most significant contributions to the Hirshfeld surfaces come from hydrogen-hydrogen (H···H) contacts, accounting for 49.7% of the interactions. Oxygen-hydrogen (O···H/H···O) and nitrogen-hydrogen (N···H/H···N) interactions, making up 12.8% and 12.7%, respectively, are represented by blue spikes on the left side, top, and bottom of the plot. The carbon-hydrogen (C···H/H···C) interactions, contributing 13.2% to the total, are placed above the O–H regions. Figure 7 displays the entire fingerprint region along with all other interactions, which are combinations of de and di.

**Table 3.** Weak hydrogen bond intermolecular interactions for $C_{16}H_{12}N_4O_2$, (I) (Å and °).

| D H A | d(D-H)/Å | d(H-A)/Å | d(D-A)/Å | D-H-A/° | Symmetry Codes |
|---|---|---|---|---|---|
| N1-H1···N4 | 0.88 | 2.19 | 3.058(2) | 169 | 1-X,2-Y,1-Z |

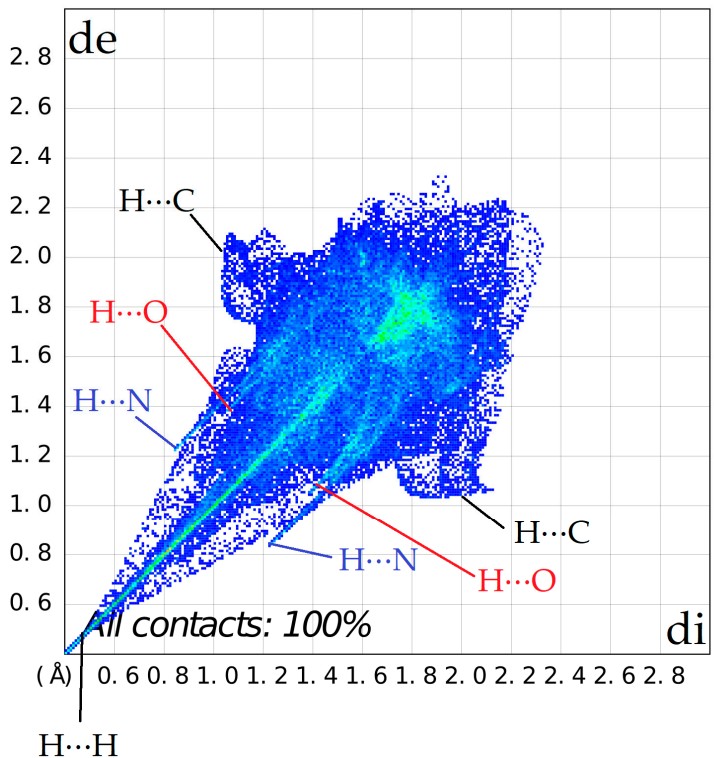

**Figure 6.** Two-dimensional fingerprint plot for $C_{16}H_{12}N_4O_2$.

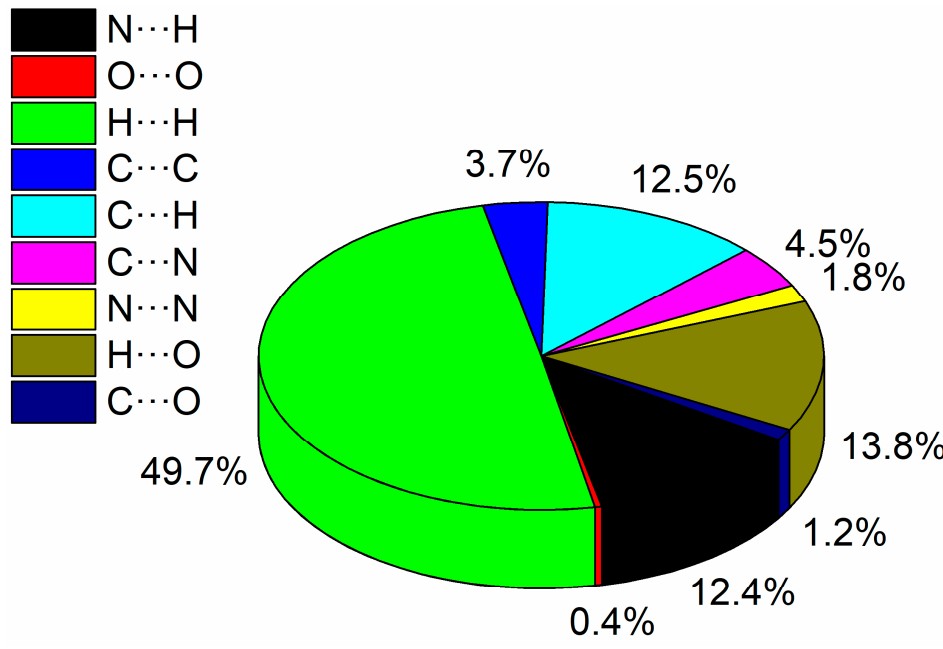

**Figure 7.** The Pie chart displays the contributions of each interaction in the title molecule.

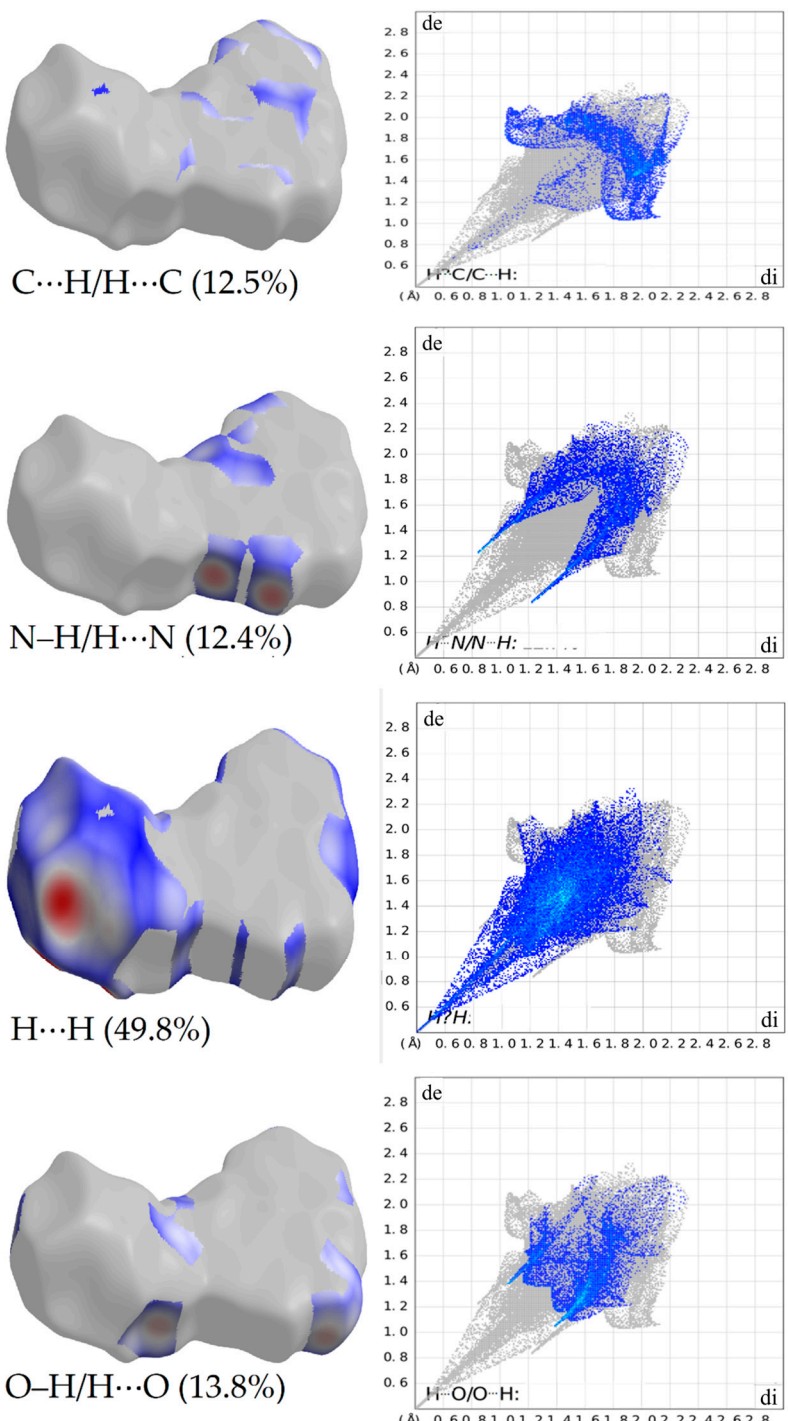

**Figure 8.** Two-dimensional fingerprint plots order with a dnorm view of the C–C (15.0%), C···H/H···C (12.5%), N–H/H···N (12.4%), H···H (49.7%), and O–H/H···O (13.8%) contacts in $C_{16}H_{12}N_4O$ Here, di and de represent separations between the nearest atoms inside and outside to the surface, respectively.

In Figure 6, the red circle on the dnorm surface of the 2-Benzyloxy-1,2,4-triazolo[1,5-a]-quinazolin-5(4H)-one structure signifies the N–H···N intermolecular interactions, as detailed in Table 2. Figure 8 illustrates the 2D&3D fingerprint, summarizing all the contacts contributing to the Hirshfeld surface. The corresponding graph for H···H showcases the 2D fingerprint of the (di, de) points associated with H atoms.

A noteworthy feature is the endpoint indicating the origin and fitting to di = de = 1.19 Å, symbolizing the occurrence of N···H/H···N contacts (12.4%), along with H···H contacts in the tested molecule (49.7%). In Figure 8, two symmetric wings on the left and right sides of the graph (13.8%) represent O···H/H···O interactions. Other interactions depicted include C···H/H···C (12.5%), and N···H/H···N (12.4%) contacts [25].

### 3.4. Interaction Energy and 3D Energy Frameworks

As indicated in prior research [9,11], the energy derived from molecular interactions within a crystalline structure can be effectively determined using the Crystal Explorer 17.5 software. We created a molecular cluster—with a radius of 3.8 Å around a selected molecule within the crystal—to calculate the total interaction energy (Figure 9). Computation of interaction energy frameworks involved the use of symmetry operations to generate molecular wave functions and to estimate the electron densities of the selected molecular cluster. The CE-B3LYP/6-31G(d,p) energy model and the following scaling factors were utilized to compute the total energy ($E_{tot}$): $K_{ele}$ = 1.057 for electrostatic interactions, $K_{pol}$ = 0.740 for polarization, $K_{dis}$ = 0.871 for dispersion, and $K_{rep}$ = 0.618 for repulsion [8,9]. The chosen basis set (CE-B3LYP/6-31G(d,p)) was employed for interaction energy calculations, as it corresponds to a higher basis set available in the latest version of the Crystal Explorer 17.5 software.

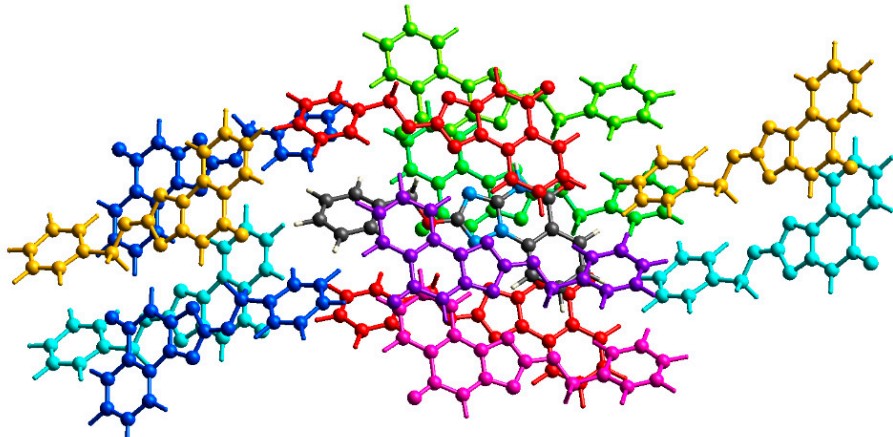

**Figure 9.** Depiction of molecular interactions between the central analyzed molecule (black color) and surrounding molecules in a cluster with a radius of 3.8 Å.

Table 4 shows the crystallographic symmetry operations and their corresponding molecular interaction energies. Here, R signifies the distance (in Å) between molecular centroids (the average atomic positions), and N represents the count of molecules at that specific distance. The energy values are reported in kJ mol$^{-1}$. A chartreuse green-colored molecule, with a symmetry operation of ($-$x, $-$y, $-$z) and stationed 6.29 Å away from the selected molecule's centroid, exhibits the maximum total interaction energy of $-$83.28 kJ mol$^{-1}$. On the other hand, a light cyan-colored molecule, bearing a symmetry operation of ($-$x + 1/2, y + 1/2, $-$z + 1/2) and positioned 14.42 Å from the selected molecule's centroid, manifests the minimum total interaction energy, recorded at $-$5.68 kJ mol$^{-1}$.

The total interaction energy, at $-$307.02 kJ mol$^{-1}$, was derived from electrostatic ($-$134.21 kJ mol$^{-1}$), polarization ($-$83.49 kJ mol$^{-1}$), dispersion ($-$288.13 kJ mol$^{-1}$), and repulsion (238.89 kJ mol$^{-1}$) components, based on molecular pair interaction energy calculations. Energy frameworks visualized in Figure 10 represent these energies using differently colored cylinders, scaled to a factor of 100 with a cutoff of $-$50 kJ/mol. Specifically, red illustrates electrostatic energy ($E_{elec}$), green shows dispersive energy ($E_{dis}$), and blue indicates total interaction energy ($E_{tot}$) (see Figure 10).

**Table 4.** Interaction energies (kJ mol$^{-1}$) for a cluster of selected molecules. For total energy ($E_{tot}$) calculation, each energy type should be multiplied by their respective factors: $K_{ele}$ = 1.057 for electrostatic, $K_{pol}$ = 0.740 for polarization, $K_{dis}$ = 0.871 for dispersion, and $K_{rep}$ = 0.618 for repulsion.

| N | Symop | R | Electron Density | $E_{ele}$ | $E_{pol}$ | $E_{dis}$ | $E_{rep}$ | $E_{tot}$ |
|---|-------|---|------------------|-----------|-----------|-----------|-----------|-----------|
| 2 | x, y, z | 5.03 | B3LYP/6-31G(d,p) | −6.7248 | −1.8926 | −51.1092 | 25.2217 | −37.4370 |
| 2 | −x + 1/2, y + 1/2, −z + 1/2 | 14.25 | B3LYP/6-31G(d,p) | −4.0063 | −1.0623 | −10.0944 | 0.0000 | −13.8121 |
| 1 | −x, −y, −z | 6.29 | B3LYP/6-31G(d,p) | −84.1110 | −20.0291 | −40.2418 | 89.8677 | −83.2813 |
| 1 | −x, −y, −z | 5.18 | B3LYP/6-31G(d,p) | −4.6088 | −2.8950 | −62.5574 | 34.0563 | −40.4533 |
| 2 | −x + 1/2, y + 1/2, −z + 1/2 | 14.42 | B3LYP/6-31G(d,p) | 1.9383 | −0.5311 | −8.4207 | 0.0000 | −5.6763 |
| 2 | x + 1/2, −y + 1/2, z + 1/2 | 13.36 | B3LYP/6-31G(d,p) | −1.6102 | −48.3443 | −19.5309 | 0.0000 | −54.4800 |
| 1 | −x, −y, −z | 4.87 | B3LYP/6-31G(d,p) | −8.3637 | −1.1691 | −56.4690 | 45.2188 | −30.9495 |
| 1 | −x, −y, −z | 6.78 | B3LYP/6-31G(d,p) | −26.7267 | −7.5670 | −39.7060 | 44.5276 | −40.9283 |
| | Summation | | | −134.21 | −83.49 | −288.13 | 238.89 | −307.02 |

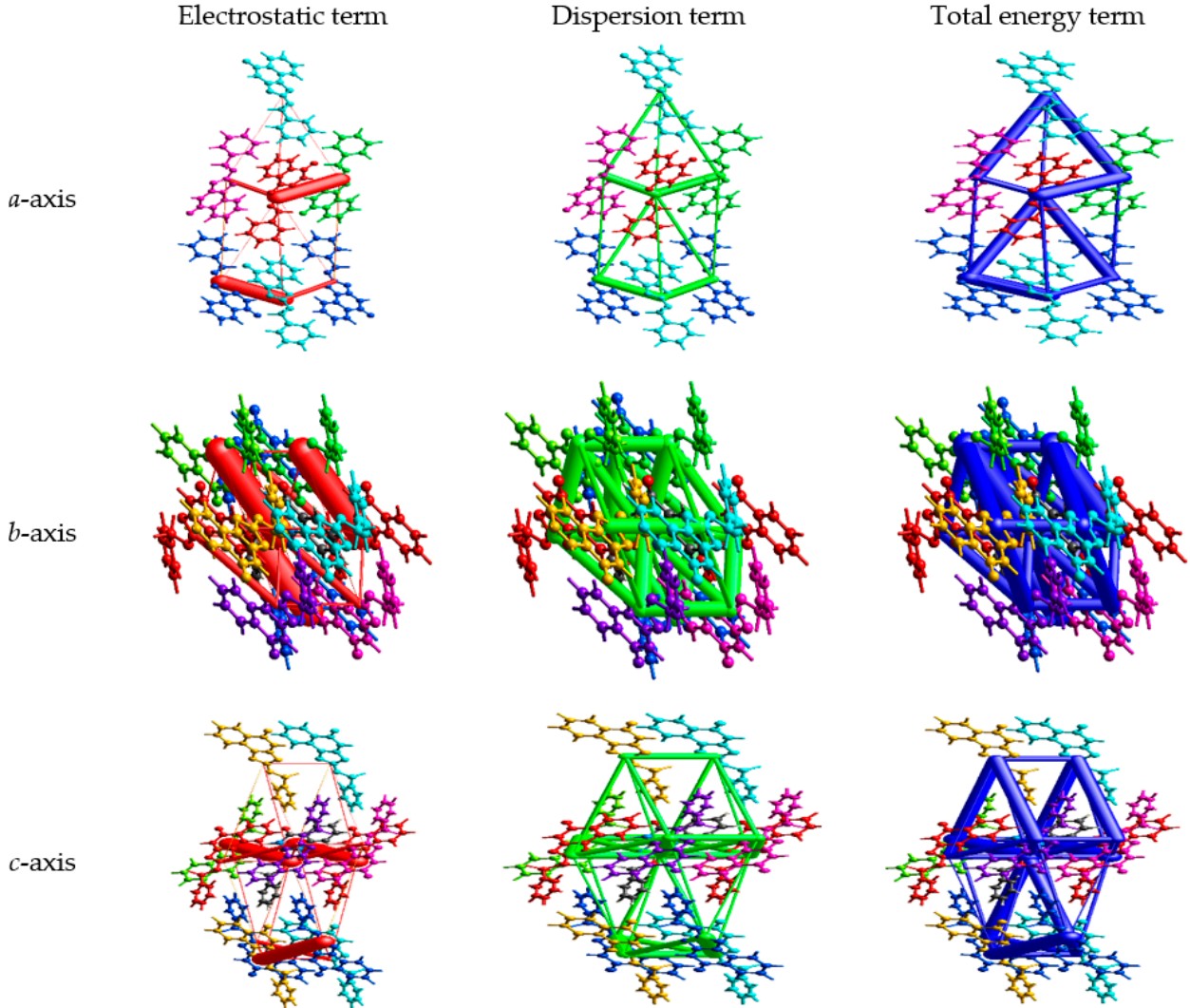

**Figure 10.** Illustration of Energy Frameworks for the Compound, Including Electrostatic Energy, Dispersion Energy, and Total Energy Components.

The computations performed within the energy framework reveal a notable dominance of dispersion energy over electrostatic and polarization energies in the crystal environment of the investigated molecule. This suggests that the distribution and correlation of electron clouds (dispersion forces) play a more substantial role in molecular interactions within this crystal than do the effects of charge distribution (electrostatic forces) or induced dipole interactions (polarization forces). The pairwise interaction energy between atoms $N_1$—$H_1 \cdots N_4$ is primarily driven by dispersion energy, measured at $-40.24$ kJ mol$^{-1}$ [30]. This amount not only exceeds the Coulomb energy but also contributes considerably to the total energy, which is $-84.11$ kJ mol$^{-1}$. This implies that the influence of the dispersion forces in this pairwise interaction is critical in the overall interaction energy within the crystal structure. When examining the interaction between stacked molecular pairs, dispersion energy again demonstrates its dominance, measured at $-51.11$ kJ mol$^{-1}$. The material's intricate arrangement and interrelations are significantly influenced by these forces. Among the variety of molecular pair interactions, it's clear that the $N_1$-$H_1 \cdots N_4$ interaction plays a crucial role in crystal packing, exerting a total energy of $-83.28$ kJ mol$^{-1}$. This underscores the importance of specific atomic interactions in shaping the overall crystal structure. Figure 11 further illustrates this point, visualizing the interaction energies involved in the formation of hydrogen bonds and molecular dimers. Lastly, the lattice energy for the polymolecule has been calculated to be $-307.02$ kJ/mol [31], a value that encapsulates the cumulative energy involved in forming the crystal lattice from isolated molecules. This is a testament to the system's overall stability and the strength of the interactions at play within the crystal structure.

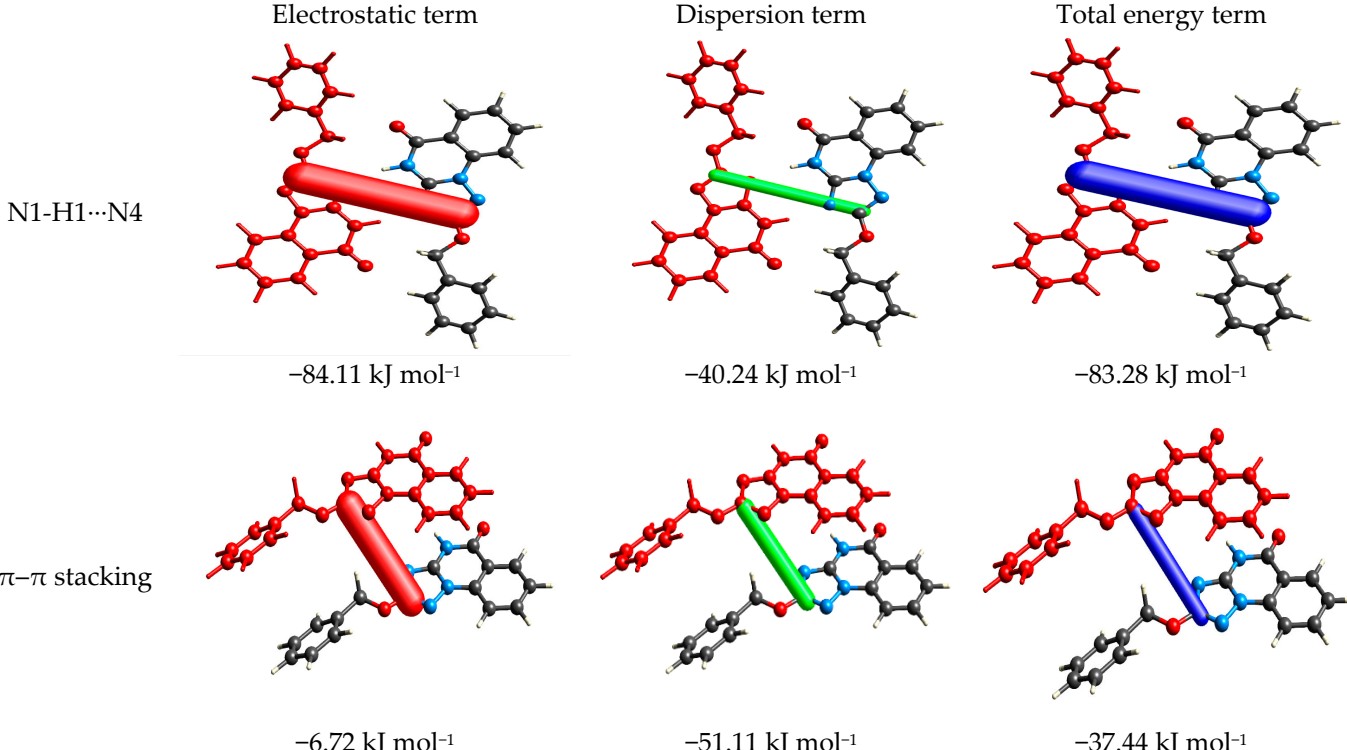

**Figure 11.** Display of interaction energies associated with hydrogen bonds and molecular dimers in the Investigated molecules.

*3.5. Frontier Molecular Orbital (FMO) Analysis*

Frontier molecular orbitals (FMO), namely the highest occupied molecular orbital (HOMO) and the lowest unoccupied molecular orbital (LUMO) are instrumental in determining various molecular properties, including chemical reactivity, stability, and optical and electrical characteristics [32]. Information pertaining to intra-molecular charge transfer can be gleaned from the analysis of these HOMO and LUMO structures. At the B3LYP/6-

31G+(d,p) level of theory, the energy levels and spatial distributions of the HOMO-5, HOMO-4, HOMO-3, HOMO-2, HOMO-1, HOMO, LUMO, LUMO+2, LUMO+1, and LUMO+3 orbitals were computed and are presented in Figure 12 [33,34].

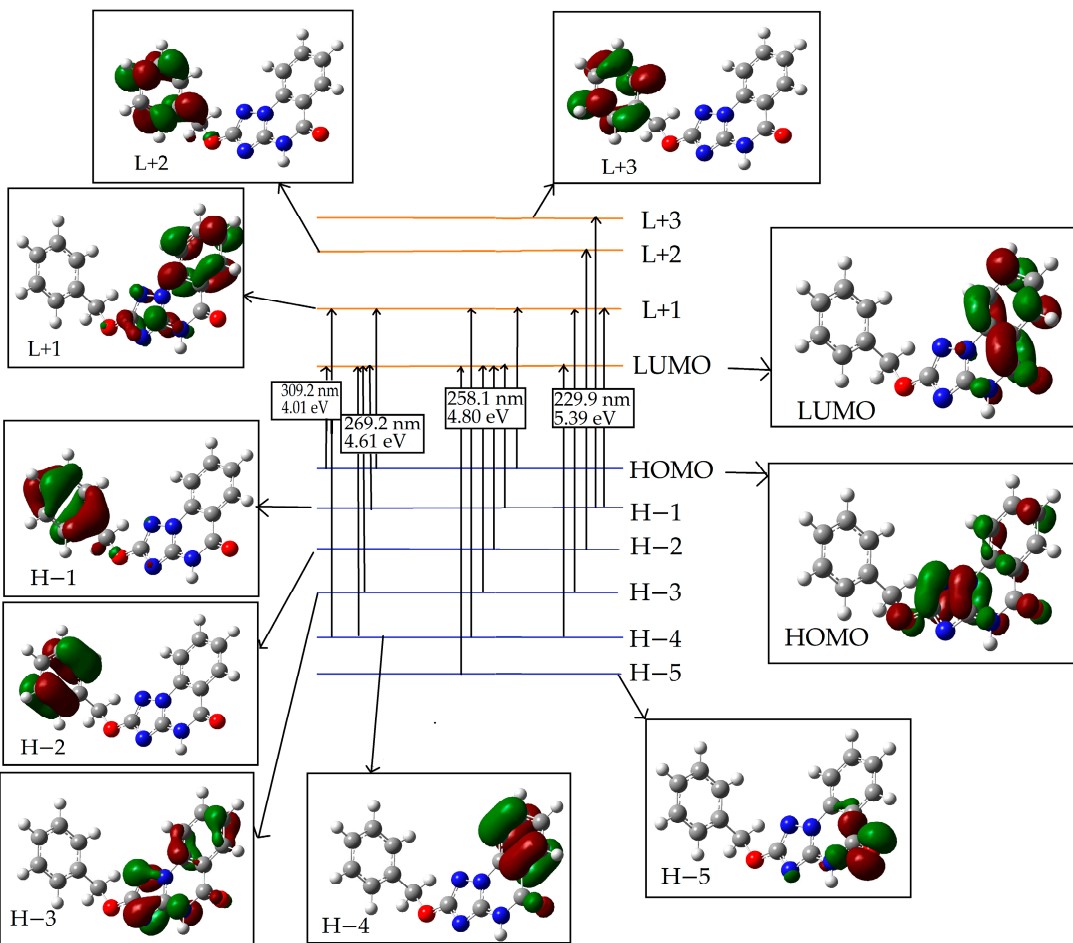

**Figure 12.** Depiction of the Frontier molecular orbitals of the Analyzed Compounds.

The frontier molecular orbital (FMO) analysis of the title compound, 2-benzyloxy-1,2,4-triazolo[1,5-a]-quinazoline, offers a valuable understanding of its electronic and optical characteristics. The compound has been identified as electron-rich, exhibiting strong electron-donating capacity in its highest occupied molecular orbitals (HOMOs) and electron-accepting potential in its lowest unoccupied molecular orbitals (LUMOs) Figure 12. This implies the compound's promising reactivity in electron transfer-involved chemical reactions.

The electron transition of the title compound, in its optimized form, was theoretically calculated in the gaseous state using the TD-B3LYP/6-311G(d,p) level of theory [35–38]. The molecular orbitals associated with the electron transition (absorption spectrum) are depicted in Figure 12. Data derived using the TD-DFT approach are tabulated in Table 5. The peak absorption for the title compound is found at 269 nm, exhibiting an oscillator strength (f) of 0. Notably, the primary electronic transition is characterized by the (HOMO)→(LUMO) transition, accounting for a substantial 96% major contribution to the overall spectral profile, while the H-4→L + 1 transition plays a minor role with a 3% contribution. This electronic excitation at 269 nm (f = 0.0944) encompasses transitions represented by four configurations: (HOMO→L + 1), (H-4→LUMO), (H-3→LUMO), and (HOMO-1→LUMO), which exhibit bandgap energies of 5.2 eV, 5.93 eV, 5.93 eV, and 5.24 eV, respectively. Conversely, the excitation at 258.1 nm (f = 0.0315) is characterized by a set of transitions described by six configurations: (H-3→LUMO), (H-1→LUMO), (H-5→LUMO),

(H-4→L + 1), (H-2→LUMO), and (HOMO→L + 1), with respective bandgap energies of 5.93 eV, 5.24 eV, 6.03 eV, 6.57 eV, 5.35 eV, and 5.2 eV [39].

**Table 5.** Presented in this table are the calculated energy values, maximum wavelengths (λ_max), Transition Energy (eV), and oscillator strengths (*f*) for the compound under investigation in its excited states in the gas phase. Also included are the identifications of electronic transitions (from the Highest Occupied Molecular Orbital (HOMO) to the Lowest Unoccupied Molecular Orbital (LUMO)), along with their primary contributions (%).

| No. | Energy (eV) | Wavelength (nm) | Osc. Strength | Symmetry | Major Contributions | Minor Contributions |
|-----|-------------|-----------------|---------------|----------|---------------------|---------------------|
| 1 | 4.01 | 309.2 | 0.0766 | Singlet-A | HOMO→LUMO (96%) | H-4→L + 1 (3%) |
| 2 | 4.61 | 269.2 | 0.0944 | Singlet-A | HOMO→L + 1 (83%) | H-4→LUMO (4%), H-3→LUMO (8%), H-1→LUMO (3%) |
| 4 | 4.80 | 258.1 | 0.0315 | Singlet-A | H-3→LUMO (25%), H-1→LUMO (56%) | H-5→LUMO (5%), H-4→L + 1 (3%), H-2→LUMO (2%), HOMO→L + 1 (7%) |
| 10 | 5.39 | 229.9 | 0.0223 | Singlet-A | H-1→L + 1 (67%) | H-4→LUMO (7%), H-3→L + 1 (7%), H-2→L + 2 (7%), H-1→L + 3 (7%) |

### 3.6. Global Reactivity Descriptors

The title compound's reactivity was probed by examining its global reactivity descriptors. These descriptors were computed from the electronic structure calculations carried out using the Gaussian 09 software package. Within these calculations, the Density Functional Theory (DFT) method with the B3LYP functional was employed. As for the basis set, the 6-311G(d,p) was utilized to provide a balance between computational accuracy and efficiency. The output generated by the Gaussian program was then analyzed to extract the relevant reactivity descriptors.

The Global Reactivity Descriptors for 2-benzyloxy-1,2,4-triazolo[1,5-a]quinazoline provides a comprehensive understanding of the compound's reactivity and stability. These indices, derived from the HOMO/LUMO band gap (ΔE), include chemical hardness (η), chemical softness (σ), global electrophilicity (ω), electronegativity (X), ionization energy (IP), and electron affinity (EA) Table 5.

$$IP = -E_{HOMO} \tag{3}$$

$$EA = -E_{LUMO} \tag{4}$$

$$X = \frac{\lfloor IP + EA \rfloor}{2} \tag{5}$$

$$\eta = \frac{\lfloor IP - EA \rfloor}{2} \tag{6}$$

$$\mu = \frac{E_{HOMO} + E_{LUMO}}{2} \tag{7}$$

$$\sigma = \frac{1}{2\eta} \tag{8}$$

$$\omega = \frac{\mu^2}{2\eta} \tag{9}$$

The ionization potential (IP) is a measure of the energy required to remove an electron from the HOMO. For the compound, the IP is 6.328, indicating a significant ability to donate electrons, given the high energy required to ionize it (Table 6). The electron affinity (EA) represents the energy change when an electron is added to the LUMO. An EA of 1.757 for this compound indicates a decent capability to accept electrons. Electronegativity (X), which measures how strongly atoms attract electrons towards them, is calculated as an average of the IP and EA. With a value of 4.0425, the compound shows a considerable tendency to attract electrons. Chemical hardness ($\eta$) and chemical potential ($\mu$) offer insights into the stability of the compound. The chemical hardness of 2.2855 signifies the compound's resistance to change in its electronic configuration, indicating considerable stability. The chemical potential, at $-4.0425$, is a measure of the compound's intrinsic tendency to exchange electrons with the environment, and this negative value shows that the compound has a low tendency to lose or gain electrons. Chemical softness ($\sigma$) is the inverse of hardness and indicates the ease of deformation of the electron cloud. The compound's softness value of 0.21877051 suggests it is relatively hard, corroborating the aforementioned hardness. The global electrophilicity index ($\omega$) measures the propensity of the compound to accept electrons. Given the $\omega$ value of 3.575105283, this compound demonstrates significant electrophilicity.

**Table 6.** Overview of global reactivity indicators for the studied compound.

| Parameters | Value (eV) |
|:---:|:---:|
| $E_{HOMO}$ | $-6.328$ |
| $E_{LUMO}$ | $-1.757$ |
| band gap ($\Delta E$) | 4.571 |
| ionization energy (IP) | 6.328 |
| electron affinity (EA) | 1.757 |
| electronegativity (X) | 4.0425 |
| chemical hardness ($\eta$) | 2.2855 |
| chemical potential ($\mu$) | $-4.0425$ |
| chemical softness ($\sigma$) | 0.218771 |
| global electrophilicity ($\omega$) | 3.575105 |

### 3.7. The Molecular Electrostatic Potential (MEP)

The reactive regions for nucleophilic (electron-poor region) and electrophilic (electron-rich region) interactions with the title compound were determined through calculations of the electrostatic potential and molecular electrostatic potential (MEP) maps [40,41]. These calculations were performed utilizing the optimized geometry at the B3LYP/6-311G(d,p) level of theory. The three-dimensional (3D) visualizations of the electrostatic potential and MEP specific to the title compound are depicted in Figures 8 and 9, respectively.

The Molecular Electrostatic Potential (MEP) map, as depicted in Figure 13, gives an illustrative depiction of the charge distribution across the surface of the compound. This map, which is color-scaled from $-5.062 \times 10^{-2}$ atomic units (au) (represented by the deepest shade of red) to $+5.062 \times 10^{-2}$ au (portrayed by the deepest shade of blue), enables a comprehensive understanding of the compound's electrostatic properties. The MEP map allows for the identification of chemical reactivity sites on the molecule's surface, which is characterized by distinct colored regions. Blue regions, which signify positive MEP values, correspond to areas of the molecule that exhibit electrophilic behavior due to an electron deficiency. These regions tend to attract nucleophiles, which are electron-rich entities.

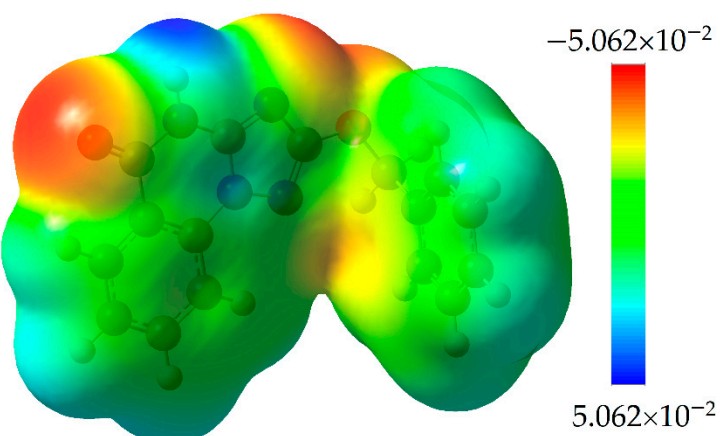

$$-5.062 \times 10^{-2}$$

$$5.062 \times 10^{-2}$$

**Figure 13.** Color Scaled Molecular Electrostatic Potential Map of the Molecule, Ranging from $-5.062 \times 10^{-2}$ au to $+5.062 \times 10^{-2}$ au.

Contrastingly, the red regions, which are associated with negative MEP values, correspond to nucleophilic areas of the molecule. These areas are electron-dense and are prone to donate electrons, thereby attracting electrophiles. The green regions, however, relate to parts of the molecule where the electrostatic potential is neutral, indicating a balanced charge distribution. These areas typically exhibit neither a significant excess nor a deficiency of electrons. Upon scrutinizing the MEP map, it becomes evident that the nucleophilic (negative) regions predominantly envelop the oxygen atoms of the molecule, indicating that these atoms are potential sites for electrophilic attack. Correspondingly, the electrophilic (blue) regions are primarily located around the hydrogen atoms that are bonded to the nitrogen atoms in the quinazolin-5(4H)-one groups of the molecule. This suggests these hydrogen atoms are likely sites for nucleophilic attack. This deeper understanding of the MEP provides valuable insights into the potential reactive sites and interactions within the molecule, which can be crucial for both experimental and computational investigations.

## 4. Conclusions

The detailed exploration conducted on the molecule 2-benzyloxy-1,2,4-triazolo[1,5-*a*] quinazolin-5-one has resulted in a profound understanding of its crystal structure, molecular geometry, molecular interactions, electronic characteristics, and potential applications. The study meticulously investigated the crystal structure, including unit cell parameters, unit cell volume, and crystal density, and validated the accuracy of the data through the quality fit between the observed and calculated electron density. The molecule's geometry optimization showed a preference for a planar conformation, a common attribute of aromatic and heterocyclic systems. The planarity of the molecule has far-reaching implications for its reactivity and molecular interactions, making it potentially significant in applications like drug-receptor interactions. The theoretical results from the DFT optimization method were in good agreement with the experimental data from single-crystal X-ray diffraction studies, further reinforcing the reliability and accuracy of the theoretical calculations.

The Hirshfeld surface analysis presented in the study provided additional insights into the intermolecular interactions within the crystal state of the compound. The analysis shed light on specific atom pair close contacts and differentiated the contributions from different interaction types. Hydrogen-hydrogen (H···H) contacts were found to be the primary contributors to these interactions. The study also demonstrated the dominance of dispersion forces in the crystal environment of the molecule by examining the interaction energy and 3D energy frameworks. The pairwise interaction energy between atoms N1—H1···N4 was primarily driven by dispersion energy, indicating its significant role in the overall interaction energy within the crystal structure.

The FMO analysis revealed the compound as electron-rich with a strong electron-donating capacity, offering valuable insights into its electronic and optical characteristics.

The band gap of 4.571 eV, the calculated maximum wavelengths, and oscillator strengths indicated the compound's potential for nonlinear optical (NLO) applications and highlighted the influence of the compound's extended conjugation on its electronic and optical properties. The ionization potential and electron affinity of the compound showcased its significant ability to donate and accept electrons, respectively. The compound's considerable stability was reflected in the values of chemical hardness and chemical potential, and the global electrophilicity index emphasized the compound's significant electrophilicity. The MEP) map visually depicted the charge distribution across the compound's surface, enabling the identification of chemical reactivity sites. The map's red and blue regions corresponded to nucleophilic and electrophilic areas of the molecule, respectively, offering insights into potential sites for electrophilic and nucleophilic attacks.

**Author Contributions:** Writing—original draft, data curation, and investigation, A.H.B.; writing—reviewing, H.A.A.; writing—reviewing and funding acquisition R.A.-S. All authors have read and agreed to the published version of the manuscript.

**Funding:** This research was funded by Researchers Supporting Project, King Saud University, Riyadh, Saudi Arabia, via project no. (RSP-2023R353).

**Data Availability Statement:** Not applicable.

**Acknowledgments:** The authors extend their appreciation to the Researchers Supporting Project, King Saud University, Riyadh, Saudi Arabia, for funding this work through grant no. (RSP-2023R353).

**Conflicts of Interest:** The authors declare that they have no known competing financial interests or personal relationships that could have appeared to influence the work reported in this paper.

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
