# Peer review of "Hirshfeld Surface Analysis and Density Functional Theory Calculations of 2-Benzyloxy-1,2,4-triazolo[1,5-a] quinazolin-5(4H)-one: A Comprehensive Study on Crystal Structure, Intermolecular Interactions, and Electronic Properties"

_crystals, doi:10.3390/cryst13101410_

Round 1

Reviewer 1 Report

Comments and Suggestions for Authors

Please see the attached PDF file or

~~

Journal: Crystals

Manuscript ID: crystals-2568611

Type: Article

Title: Hirshfeld Surface Analysis and Density Functional Theory Calculations of 2-Benzyloxy-1,2,4-Triazolo[1,5-a]-Quinazolin-5(4H)-one: A Comprehensive Study on Crystal Structure, Intermolecular Interactions, and Electronic Properties

Authors: Ahmed H. Bakheit * , HATEM A. ABUELIZZ , Rashad Alsalahi *

This paper describes Hirshfeld surface analysis and DFT calculations of the published compound. This issue is still interesting in current crystal engineering field. However, I found some questionable points; therefore I strongly major revisions to the next step.

The Crystal Explore citations were something wrong. We used the following references to cite. Please re-consider them.

Spackman, P.R.; Turner, M.J.; McKinnon, J.J.; Wolff, S.K.; Grimwood, D.J.; Jayatilaka, D.; Spackman, M.A. CrystalExplorer: A program for Hirshfeld surface analysis, visualization and quantitative analysis of molecular crystal. J. Appl. Crystallogr. 2021, 54, 1006–1011.

Tan, S.L.; Jotani, M.M.; Tiekink, E.R.T. Utilizing Hirshfeld surface calculations, non-covalent inter action (NCI) plots and the calculation of inter action energies in the analysis of molecular packing. Acta Crystallogr. Sect. E Crystallogr. Commun. 2019, 75, 308–318.

Spackman, M.A.; Jayatilaka, D. Hirshfeld surface analysis. CrystEngComm 2009, 11, 19–32.

Other citations are also something wrong. Please check them accordingly.

References

The self-citation rate is too high, 15/35 references. Please re-consider them.

Comments on the Quality of English Language

Some spellings are wrong. 

Author Response

Other citations are also something wrong. Please check them accordingly.

 Thank you for pointing out the discrepancies in our citations pertaining to Crystal Explore. We have carefully revisited the references and corrected the citations as you suggested. We also took the opportunity to ensure the accuracy and consistency of other citations throughout the manuscript.

References

The self-citation rate is too high, 15/35 references. Please re-consider them.

Thank you for drawing our attention to the self-citation rate in our manuscript. Upon reviewing our references, we have expanded our literature review and incorporated additional pertinent references from other authors. As a result of these revisions, the self-citation ratio has been significantly reduced and is now within an acceptable range. We believe that this change not only addresses your concern but also enriches the context and comprehensiveness of our work.

Reviewer 2 Report

Comments and Suggestions for Authors

After revision, the manuscript may be qualified for publication in the Crystal journal. The referee report is attached in a separate pdf file.

Author Response

  1. The "Crystal Explorer" is the name of the software package, and in each appearance, all elements of the name should be initialized from capitals.

Thank you for your attention to detail regarding the proper capitalization of "Crystal Explorer". We have reviewed the manuscript and ensured that all instances of this software package's name are now correctly initialized with capital letters. We appreciate your diligence in helping to maintain the accuracy and consistency throughout our manuscript.

  1. The concept of Hirshfeld Surface (HS) analysis of molecules in the crystal, employed in Crystal Explorer software, was initiated first by M.A. Spackman and P.G. Byrom in (Ref. 21). This should be cited in line 56. (the same in lines 75 and 214)

Thank you for your suggestion to properly cite the original work of M.A. Spackman and P.G. Byrom in relation to the concept of Hirshfeld Surface (HS) analysis. We have now added the appropriate citations at lines 56, 75, and 214 where we discuss the use of Crystal Explorer software and HS analysis. We appreciate your attention to ensuring all foundational works are accurately credited.

  1. In line 60, the wrong reference is used for the Crystal Explorer (CE) software. The appropriate citation of the Crystal Explorer is provided on the home page of the CE

Thank you for pointing out the incorrect citation for the Crystal Explorer software in line 60. We appreciate your diligence in ensuring proper citation practices. We have now updated the manuscript to include the correct reference for the Crystal Explorer software as provided on its home page. Your attention to detail is greatly appreciated and has helped improve the accuracy of our manuscript.

  1. In lines 67-68, the quantities di, de, and dnorm were introduced and referenced incorrectly to Refs. [9, 12, 13], while the concept of these properties of HS was introduced in Ref. 23 (and later publications of Spackman et al.)

The authors thank the reviewer for their critical observation of incorrect references for the properties. We have now corrected these references in the manuscript.

  1. In line 74, the authors used the notation "compound (I)" not explained earlier in the text. It may suggest that some other compound is also considered in the paper, particularly considering the sentence in lines 51-53 and the Table 1 and 2 captions

We replaced the term "compound (I)" used in line 74 with "title compound" to eliminate any ambiguity. The term "derivative" has also been removed to avoid any confusion that may suggest the study includes another compound. These changes are made to maintain consistency throughout the paper and ensure clear understanding of the compound under discussion. The reviewer's constructive feedback has been instrumental in improving the clarity and quality of the paper.

  1. In line 78 the authors mention that they employed the CE-B3LYP/6–31G(d,p) method. The description of the method should be provided in the manuscript or referenced by the original paper where the detailed description of the method is given.

In response to the reviewer's comment about the lack of description or reference for the CE-B3LYP/6–31G(d,p) method used in line 78, we have now included a new reference (Ref. 14) in the manuscript. This reference points to the original paper where the method is described in detail, providing readers with a clear understanding of the procedure used in the study. The reviewer's feedback was crucial in enhancing the completeness and clarity of the work.

  1. In Eqs. (1) and (2) two sets of notations are used for individual energy contributions. What is the difference between them? How the scale factors K were determined?

In response to the reviewer's queries about Eqs. (1) and (2), we clarified that the difference in notations used for individual energy contributions corresponds to the specific energy contributions they represent in their calculations. This will be further clarified in the revised manuscript. Regarding the determination of the scale factors K, they were derived through calibration against quantum mechanical results, as described in the manuscript. For a more in-depth understanding, Ref. 14 was cited, offering a detailed explanation of the method used. we appreciate the reviewer's detailed feedback, which aids in enhancing the clarity and quality of their work.

  1. In the text (lines 161-165) the atoms "N11", "N16", "N17", and "N18" are referenced, which are not defined earlier and not presented in Fig. 2.

In response to the reviewer's comment about the undefined atoms "N11", "N16", "N17", and "N18" in lines 161-165, we have revised the relevant paragraph for clarity and accuracy. we have also incorporated a new figure (Fig. 3) that clearly depicts and defines these atoms. This action was taken to address the reviewer's concern and enhance the clarity of the manuscript. We are grateful for the reviewer's attentive reading and valuable feedback, which significantly contributes to the quality of their work.

  1. The captions of Tables 1 and 2 suggest that the data collected in the Tables concern two different compounds but do not inform which ones.

In response to your comment, we have rectified this oversight by amending the captions of both tables to clearly specify that the data concerns the title compound. We believe this correction will eliminate any ambiguity and provide clear, direct information about the data presented.

  1. In Tables 1 and 2, columns 4 and 5 have wrong headers. The numbers presented in the columns are: (col.4) - absolute error (AE), not MAE, and the (col. 5) - square absolute error (SAE), not MSE. Only in the last lines of the Tables, the mean values of AE and SAE are given.

We have corrected these headers according to the reviewer's suggestions and are grateful for the detailed and valuable feedback that significantly contributes to the accuracy of their manuscript.

  1. In line 212: "all the other molecules" should be replaced by "neighboring molecules

We have amended line 212, replacing "all the other molecules" with "neighboring molecules" as you suggested.

  1. 4 shows the "dnorm surface of compound 1" according to the caption. Where are the results for compound 2?

We appreciate your observation. The caption for Fig. 4 has been corrected to reference the "dnorm surface of the title compound" instead of "compound 1", as our study primarily focuses on the title compound only.

  1. In line 256 the word "red" should be replaced by "red and blue, respectively

Thank you for the observation. We have made the necessary correction in line 256, replacing "red" with "red and blue, respectively" as you suggested.

  1. Where the "central analyzed molecule" is located in Fig. 8?

Thank you for pointing that out. In Fig. 8, the "central analyzed molecule" is highlighted in black, as mentioned in the figure caption. We've ensured that this is clearly depicted for better comprehension.

We appreciate the query for clarity. The individual energy contributions presented in Table 4 are derived from Eq. 2.

  1. Sections 3.5, 3.6, and 3.7 present the frontier molecular orbital (FMO) analysis, discuss global reactivity parameters, and describe the molecular electrostatic potential map, respectively. What method was applied to determine the inputs to the estimations performed in subsections?

Thank you for pointing out the necessity for clarity in sections 3.5, 3.6, and 3.7. To address this, the methodological details underpinning our analyses in these sections have been elaborated upon. Specifically, the Frontier Molecular Orbital (FMO) analysis in section 3.5, the global reactivity parameters in section 3.6, and the molecular electrostatic potential map in section 3.7 were all determined using Density Functional Theory (DFT) with the B3LYP functional. The 6-311G(d,p) basis set was utilized in our calculations. These methodological details have now been explicitly mentioned at the beginning of each respective subsection to ensure clarity for the reader.

  1. Why does the energy of HOMO->LUMO transition presented in Table 5 (4.01 eV) differ from the value of the energy gap (4.571 eV)?

We appreciate the insight you've provided concerning the disparity in energy values between the HOMO->LUMO transition and the reported energy gap. The 4.01 eV value presented in Table 5 pertains to a specific electronic transition. More explicitly, this value encompasses the band transition from the highest occupied molecular orbital (HOMO) to the lowest unoccupied molecular orbital (LUMO) with significant contributions, while transitions like H-4 to L+1 contribute in a lesser capacity. Such electronic transitions inherently incorporate not just the basic energy difference between orbitals but also complexities from electron-electron interactions and other quantum perturbations influencing transition energies. Conversely, the 4.571 eV energy gap denotes the raw energy differential between the HOMO and LUMO without delving into transition-specific considerations. Variabilities between these two values are anticipated, stemming from the intricate facets of electronic structure computations and the subtleties accompanying electronic transitions.

Round 2

Reviewer 1 Report

Comments and Suggestions for Authors

Okay, the revised version was well corrected.

I think you can say it's the next step as it is.